# Growth, Toxin Content and Production of Dinophysis Norvegica in Cultured Strains Isolated from Funka Bay (Japan)

**DOI:** 10.3390/toxins15050318

**Published:** 2023-05-01

**Authors:** Satoshi Nagai, Leila Basti, Hajime Uchida, Takanori Kuribayashi, Masafumi Natsuike, Sirje Sildever, Natsuko Nakayama, Wai Mun Lum, Ryuji Matsushima

**Affiliations:** 1Coastal and Inland Fisheries Division, Fisheries Technology Institute, Japan Fisheries Research and Education Agency, 2-12-4 Fukuura, Kanazawa-ku, Yokohama 236-8648, Japan; 2Department of Integrative Agriculture, College of Agriculture and Veterinary Medicine, United Arab Emirates University, Al Ain P.O. Box 15551, Abu Dhabi, United Arab Emirates; bastileila@gmail.com; 3Faculty of Marine Resources and Environment, Tokyo University of Marine Science and Technology, 4-5-7 Konan, Minato-ku, Tokyo 108-8477, Japan; 4Seafood Safety and Technology Division, Fisheries Technology Institute, Japan Fisheries Research and Education Agency, 2-12-4 Fukuura, Kanazawa-ku, Yokohama 236-8648, Japan; uchida_hajime03@fra.go.jp (H.U.);; 5Headquarters, Hokkaido Research Organization, Kita 19 Nishi 11, Kita-ku, Sapporo 060-0819, Japan; kuribayashi-takanori@hro.or.jp; 6Hakodate Fisheries Research Institute, Hokkaido Research Organization, 20-5 Benten-cho, Hakodate 040-0051, Japan; natsuike-masafumi@hro.or.jp; 7Department of Marine Systems, Tallinn University of Technology, Akadeemia tee 15A, 12618 Tallinn, Estonia; 8Environmental Conservation Division, Fisheries Technology Institute, Japan Fisheries Research and Education Agency, 2-17-5 Maruishi, Hatsukaichi 739-0452, Japan; nakayama_natsuko37@fra.go.jp; 9Department of Aquatic Bioscience, Graduate School of Agricultural and Life Sciences, The University of Tokyo 1-1-1 Yayoi, Bunkyo, Tokyo 113-8657, Japan; lumwaimun@gmail.com

**Keywords:** diarrhetic shellfish poisoning (DSP), *Dinophysis norvegica*, *Mesodinium rubrum*, *Teleaulax amphioxeia*, pectenotoxin-2 (PTX2), dinophysistoxin-1 (DTX1), okadaic acid (OA)

## Abstract

The successful cultivation of *Dinophysis norvegica* Claparède & Lachmann, 1859, isolated from Japanese coastal waters, is presented in this study, which also includes an examination of its toxin content and production for the first time. Maintaining the strains at a high abundance (>2000 cells per mL^−1^) for more than 20 months was achieved by feeding them with the ciliate *Mesodinium rubrum* Lohmann, 1908, along with the addition of the cryptophyte *Teleaulax amphioxeia* (W.Conrad) D.R.A.Hill, 1992. Toxin production was examined using seven established strains. At the end of the one-month incubation period, the total amounts of pectenotoxin-2 (PTX2) and dinophysistoxin-1 (DTX1) ranged between 132.0 and 375.0 ng per mL^−1^ (n = 7), and 0.7 and 3.6 ng per mL^−1^ (n = 3), respectively. Furthermore, only one strain was found to contain a trace level of okadaic acid (OA). Similarly, the cell quota of pectenotoxin-2 (PTX2) and dinophysistoxin-1 (DTX1) ranged from 60.6 to 152.4 pg per cell^−1^ (n = 7) and 0.5 to 1.2 pg per cell^−1^ (n = 3), respectively. The results of this study indicate that toxin production in this species is subject to variation depending on the strain. According to the growth experiment, *D. norvegica* exhibited a long lag phase, as suggested by the slow growth observed during the first 12 days. In the growth experiment, *D. norvegica* grew very slowly for the first 12 days, suggesting they had a long lag phase. However, after that, they grew exponentially, with a maximum growth rate of 0.56 divisions per day (during Days 24–27), reaching a maximum concentration of 3000 cells per mL^−1^ at the end of the incubation (Day 36). In the toxin production study, the concentration of DTX1 and PTX2 increased following their vegetative growth, but the toxin production still increased exponentially on Day 36 (1.3 ng per mL^−1^ and 154.7 ng per mL^−1^ of DTX1 and PTX2, respectively). The concentration of OA remained below detectable levels (≤0.010 ng per mL^−1^) during the 36-day incubation period, with the exception of Day 6. This study presents new information on the toxin production and content of *D. norvegica*, as well as insights into the maintenance and culturing of this species.

## 1. Introduction

Harmful algal blooms (HABs) are caused by several species of microalgae in freshwater, marine, and brackish environments. These events may lead to important impacts on the ecosystem and on a socioeconomic level, as well as human illnesses [1,2,3]. There has been a marked increase in the occurrence of HABs worldwide [1,2,3,4,5]. The rise has been associated with climate change and intensified anthropogenic activities, notably, eutrophication, transport of species with maritime activities, alteration of natural habitats, and growth of the aquaculture industry [4,6,7,8,9,10,11,12,13,14].

In marine waters, dinoflagellates include the majority of toxin-producing HAB species and are responsible for several human poisoning syndromes, including Ciguatera fish poisoning (CFP), neurotoxic shellfish poisoning (NSP), paralytic shellfish poisoning (PSP), and diarrhetic shellfish poisoning (DSP) [15]. For example, gastrointestinal poisoning in humans is caused by the consumption of shellfish contaminated with DSP toxins [16,17]. These toxins are produced by dinoflagellates from the genera *Dinophysis*, *Phalacroma,* and *Prorocentrum*. Ten species of *Dinophysis* and two species of *Phalacroma* are known to produce lipophilic diarrhetic shellfish toxins, or DSTs, i.e., okadaic acid (OA) and its analogs such as dinophysistoxins (DTX), principally DTX1, DTX2, and DTX3, in addition to bioactive pectenotoxins, PTX [17,18,19,20,21,22,23,24,25,26,27,28].

Typically, *Dinophysis* spp. do not attain high cell densities, but form dense patches of populations, which sets them apart from other HAB species and makes their monitoring and prediction of shellfish contaminations with DSTs more difficult, especially since molecular tools have been hard to develop due to insufficient resolution to differentiate between the species [21,29,30,31,32,33,34,35,36].

Despite the availability of extensive studies, little is known about the ecophysiology, bloom mechanisms, and toxin production of the *Dinophysis* species due to difficulties in establishing and maintaining cultures [21,37,38,39]. The discovery of mixotrophy in *Dinophysis* species [37,38,39,40] and plastids of cryptophyte origin [40,41,42,43,44,45,46,47] led to the first success in establishing cultures of *Dinophysis acuminata* Claparède & Lachmann, 1859 [48]. Seven species were subsequently cultured based on feeding *Dinophysis* spp. with the ciliate *Mesodinium rubrum* grown with the cryptophyte *Teleaulax* sp., namely, *D. fortii* Pavillard, 1924 [49]; *D*. *acuta* Ehrenberg, 1839 [50]; *D. sacculus* F. Stein, 1883 [51]; *D. tripos* [52]; *D*. cf. *ovum* (F.Schütt) T.H.Abé [53]; *D. caudata* Saville-Kent, 1881 [54]; and *D. infundibulum* J. Schiller, 1928 [55]. Mainly growth and, in some cases, toxin production in the established cultures have been reported. A few studies have investigated the effects of temperature, prey, and irradiance on the growth and toxin production of *Dinophysis* spp. in these cultures [33,48,56,57,58,59,60,61,62,63,64,65].

The global expansion of *Dinophysis* species associated with climate change and aquaculture activities has resulted in difficulties for fisheries and the aquaculture industry through the long-term closure of shellfish-producing areas [14,66,67]. Among the toxigenic species, six of the *Dinophysis* species are widely distributed across the globe, including *D*. *norvegica* [36,68,69]. *D. norvegica* is usually reported in the northern hemisphere, which is the cold-temperature region, for example, including the coastal waters around Scotland and Norway, the Baltic Sea, and the Arctic Sea [29,30,70,71,72,73,74,75,76]. Recently, the species was reported for the first time at very low occurrence in oceanic samples in the southern hemisphere, in the southern Argentine Sea [77]. It forms dense blooms in the Baltic Sea and eastern Canada, with mild DSP outbreaks [72,78,79,80]. The earliest information from 1989 based on cells isolated from field samples showed the production of either OA or DTX1, or both, in Japan, Norway, and Spain [17], and a high content of OA in eastern Canada [81]. More recently, LC-MS have shown the production of PTX2, PTX12, and traces of OA by strains from Norway [22]. In the Baltic Sea, *D*. *norvegica* produces OA, PTX2, and PTX2SA [82], leading to the contamination of blue mussels and flounders with OA [83,84]. One recent study reported the production of Dihydrodinophysistoxin-1 in isolated cells from environmental samples and cultures of *D*. *norvegica* from the Gulf of Maine, USA [85], with a complete absence of OA, DTX1, and DTX2 following analyses with LC-MS/MS. In Japan, high levels of PTX2 have been reported for the first time in cells of *D*. *norvegica* isolated from field samples [86]. In a later study, PTX2 was confirmed as the dominant toxin in *D*. *norvegica*, although some of the isolated cells had trace levels of OA and DTX1 [87]. In the present study, we report the successful cultivation of *D*. *norvegica* isolated from Japanese waters for the first time. The toxin productions in seven strains of *D*. *norvegica* are provided, as well as the information on the growth and toxin production of one strain (DN16062021FUN-06) during a 36-day culture experiment.

## 2. Results

### 2.1. Species Identification

*Dinophysis norvegica* cells are generally large, ovoid, and robust. The posterior end tapers to a triangular shape (Figure 1). The micrographs showed the large nucleus which occupied the upper half of the cell and food vacuoles at the lower part (Figure 1A) and numerous chloroplasts (Figure 1B). *Dinophysis acuta* closely resembles *D. norvegica* based on morphology; therefore, they may be misidentified. These species can be distinguishable by their size (although it overlaps) and the location of the widest position: *D. acuta* is larger and widest below the mid-section, whereas *D. norvegica* is smaller and widest in the middle region of the cell [88,89]. A phylogenetic analysis based on the D1/D2 region (735 bp) supported that the strains isolated from Funka Bay, Japan, belong to the *D. norvegica* clade (Figure 2), and are closely related to the strains from Canada, Norway, and the USA (Atlantic Ocean).

### 2.2. Feeding Behavior and Growth of Dinophysis Norvegica from Funka Bay in Culture Experiments

Only seven cultures of 48 single-cell isolates grew with the addition of the ciliate *Mesodinium rubrum* from the Oita Prefecture (Japan) as the prey species. However, clonal strains were successfully established, and the isolation success was 14.6% (7/48). Maximum cell densities of cultured strains during the 36 days of incubation ranged from 1057 cells per mL^−1^ to 3050 cells per mL^−1^ (mean of 2020 ± 702 cells per mL^−1^; n =7). Similar to the case of other *Dinophysis* species, *Dinophysis norvegica* was able to feed on the ciliate *M. rubrum* and grow.

In the growth experiment (12.5 °C, 12:12 light/dark cycle and irradiance of 100 μmol m^−2^ s^−1^), the ciliate *M. rubrum* grew exponentially during the first 9 days, reaching 5600 ± 346 cells per mL^−1^ (mean ± SD, n = 3) (Figure 3). The information on the experimental conditions is available in Section 4.1. After that, cell abundances of *M. rubrum* decreased sharply and disappeared by Day 30, probably due to the active consumption by *D. norvegica* and natural death. *Dinophysis norvegica* grew, but it showed very slow growth for the first 12 days, suggesting that they have a long lag phase. After that, they grew exponentially, with a maximum growth rate of 0.56 divisions per day^−1^ (from Day 24 to Day 27), reaching a maximum concentration of 2883 ± 104 cell per mL^−1^ at the end of incubation (Day 36), and the growth rates in every third day from Day 0 to Day 30 were −0.06 to 0.56 divisions per day^−1^ (0.18 ± 0.18) (Figure 3).

### 2.3. Toxin Production

Low levels of DTX1 were found, but only in cells of three strains of *D*. *norvegica* at 0.5 pg per cell^−1^ for DN16062021FUN-05, 0.7 pg per cell^−1^ for DN16062021FUN-08, and 1.2 pg per cell^−1^ for DN16062021FUN-06. All seven strains, however, produced PTX2 with cell quotas varying from 60.6 pg per cell^−1^ to 152.4 pg per cell^−1^ (Table 1). In cultures, the toxin concentration ranged from 0.705 ng per mL^−1^ to 3.55 ng per mL^−1^ in the case of DTX1 and from 126 ng per mL^−1^ to 375 ng per mL^−1^ in the case of PTX2 (Table 1). All strains of *Dinophysis norvegica* did not produce OA, except for strain DN16062021FUN-08, in which we detected a trace level of OA. Thus, the PTX2 showed clear peaks in all *D. norvegica* strains, and the DTX1 was detected in enough concentrations to be quantified from three strains, DN16062021FUN-05, DN16062021FUN-06, and DN16062021FUN-08. OA was lower than the detection limit concentration in all strains, but there was a small peak with a signal-to-noise ratio of 4 at the same retention time as OA in the MRM chromatograms of strain DN16062021FUN-08 (Figure 4). It may be possible that OA could be detected from samples with high cell density and concentration.

Strain DN16062021FUN-06 of *D. norvegica* produced DTX1 and PTX2 throughout the 36-day culture experiment (Figure 5). Both productions of DTX1 and PTX2 started to increase from 0.092 ± 0.009 pg per cell^−1^ on Day 12 to 1.280 ± 0.185 pg per cell^−1^ on Day 36, and from 9.3 ± 0.5 pg per cell^−1^ on Day 12 to 154.7 ± 24.2 pg per cell^−1^ on Day 36, respectively. The specific toxin production rates for DTX1 and PTX2 during the exponential growth phase were 0.291 ± 0.020 pg per mL^−1^ per day^−1^ and 0.291 ± 0.023 pg per mL^−1^ per day^−1^, respectively. The net toxin production rate (R_tox_) during the exponential growth phase was 0.001 ± 0.0003 ng per mL^−1^ per day^−1^ for DTX1 and 0.13 ± 0.03 ng per mL^−1^ per day^−1^ for PTX2 in the exponential phase. DTX2 concentrations that were above the detection limit were detected from Day 18 to Day 36 and ranged from 0.132 to 1.28 ng per mL^−1^. For PTX2, the concentrations above the detection limit were measured from Day 3 until the end of the experiment, Day 36, and were in the range of 3.4 to 154.7 ng per mL^−1^.

## 3. Discussion

Similar to the case of other *Dinophysis* species [49,50,51,52,53,54,55], *Dinophysis norvegica* could feed on the ciliate *M. rubrum* by inserting the peduncle into the cells. The ciliate became immobile just after insertion, and their cilia were shed from the cell within 1–5 min. The cytoplasm of the prey was actively ingested through the peduncle. It took 45–100 min until the whole cell content of *M. rubrum* was consumed by *D. fortii* and *D. tripos* [49,52]. *Dinophysis acuminata*, *D. caudata*, and *D. fortii*, isolated from Japanese coastal waters, displayed growth rates of 0.50–1.03 divisions per day^−1^, reaching maximum concentrations of 2200–11,000 cells per mL^−1^ at temperatures ranging from 18 to 25 °C [36]. In this study, *D. norvegica* showed a similar growth rate at a lower temperature (12.5 °C) than other species. The culture strains were maintained successfully at high densities (>2000 cells per mL^−1^) for more than 20 months. This is an advancement compared to previous findings, where clonal cultures of *D. norvegica* isolated from Funka Bay and Lake Notoroko in Hokkaido, Japan, grew well (>1000 cells per mL^−1^) in the first incubation (24/96, 25%). However, when they were reinoculated into fresh *Mesodinium* cultures, no further growth was confirmed, and thus, it was not possible to establish their cultures (0/96, 0%) [52]. This led to the conclusion that the predator and the prey from different regions may be incompatible and cause failure in culturing [52]. In the growth experiment using the established cultures, *D. norvegica* had a much longer log phase (12 days) (Figure 3) than *D. acuminata*, *D. caudata,* and *D. fortii* (3–4 days) [49,54,55]. *D. norvegica* may become unstable when reinoculated into fresh *Mesodinium* cultures because of sudden environmental changes such as changes in pH. It is suggested that the reinoculation of *D. norvegica* cells with a relatively high concentration (>250 cells per mL^−1^) would achieve the successful maintenance of the cultures for long periods. The ranges of water temperatures and salinity in which this species appears in southern Hokkaido, including Funka Bay, were 2–16 °C and 24.3–33.9 PSU during 2016–2020 (https://www.hro.or.jp/list/fisheries/research/central/section/kankyou/kaidoku/j12s220000000dgw.html, accessed on 2 February 2023). The influence of the temperature (12.5 °C) on the success of establishing cultures is still unclear in this study.

The establishment of strains in the toxic *Dinopshysis* species is important for investigating toxin production and understanding how it is influenced by changes in physical–chemical conditions. Until now, information on the concentration of toxins produced by *D. norvegica* has been available based on isolated and pooled cells from field samples from two locations in Norway (Sogndal and Flødevigen Bay) and in Japan (Yakumo and Saroma) [19,22,87]. Similar to the previous records based on isolated cells, OA concentrations in cultured strains remained below the detection limit [19,22,87]. The maximum DTX1 concentrations in cultured strains were lower than reported based on cells from field samples (1.16 pg per cell^−1^ vs. 14 pg per cell^−1^) [19,22,87]. Interestingly the PTX2 maximum concentrations were much higher in the cultured strains (152 pg per cell^−1^), than in the cells collected from the field in Yakumo Japan (89 pg per cell^−1^) and 1.7 pg per cell^−1^ from Flødevigen Bay in Norway [22,87]. From 1987 until 2022, 43 HABs associated with *D. norvegica* were recorded globally, the majority of them associated with DSP [90]. A study based on Scottish shellfish farms estimated that a 1% increase in the toxins produced by *Dinophysis* spp. can lead to a 0.66% reduction in shellfish production, resulting in an estimated annual loss of GBP 1.37 million [91]. Additional studies on strains in culture may therefore enhance our understanding of not only the variability in toxin production between strains from the same and different geographical locations but also the impact of changes in physicochemical variables. [69,92].

## 4. Materials and Methods

### 4.1. Isolation and Establishment of Clonal Cultures

The ciliate *Mesodinium rubrum* and cryptophyte *Teleaulax amphioxeia* were isolated from Inokushi Bay (32.7998 N, 131.8923 E) in Oita Prefecture, Japan, at the end of February 2007 [49]. To sustain the *M. rubrum* culture, a mixture of 50 mL of the culture (7.0–9.0 × 10^3^ cells per mL^−1^) and 100 mL of a modified f/2 medium [93,94] was prepared, along with 25–100 µL of *T. amphioxeia* culture (containing 0.5–2.0 × 10^4^ cells) as a food source. The culture medium was prepared based on autoclaved natural seawater collected from Tokyo Bay (35.3460 N, 139.6570 E) with the addition of 1/3 nitrate, phosphate, metals, and 1/10 vitamins. Before autoclaving, the salinity of the seawater was adjusted to 30 practical salinity units (PSU). The ciliate culture was maintained at a temperature of 18 °C under a photon irradiance of 100 μmol m^−2^ s^−1^, provided by cool-white fluorescent lamps, with a 12:12 h light/dark cycle.

Cells of *Dinophysis norvegica* (48 cells in total) were isolated by micropipetting from a seawater sample collected from Funka Bay, Japan (42.28 N, 140.35 E), in June 2021 and incubated in individual wells of a 48-well microplate (Iwaki, Japan). Each *D. norvegica* cell was grown in 1.0 mL of the culture medium, containing ca. 1.0 × 10^3^ cells of *M. rubrum* as the prey species. *Dinophysis* cells were incubated under the same light conditions as those for the *M. rubrum* culture, but at 12.5 °C. After one month of incubation, seven strains grew well and were established as clonal strains. The established cultures in each strain were maintained by inoculating small aliquots (0.1 mL) into 2.9 mL of fresh *M. rubrum* culture (approximately 2 × 10^3^ cells per mL^−1^) in 12-well microplates. These microplates were then incubated for a month under the same conditions as mentioned above, without the addition of *Teleaulax* culture. After one month of incubation, 1 mL of culture from each strain was sampled for toxin analysis. Cell abundances were estimated on 100 μL of cultures (in triplicates) inoculated and fixed with Lugol’s solution (2%) into a 96-well plate (Iwaki, Japan). Cells were counted under the inverted microscope (Nikon TE-300). Additionally, the orange autoflorescence derived from the chloroplasts of *M. rubrum* in *D. norvegica* cells was observed using an epifluorescence microscope under blue light excitation ((Zeiss Axioskop 2 (Carl Zeiss, Göttingen, Germany)) equipped with a digital camera (Axiocam 305 color (Carl Zeiss, Göttingen, Germany)).

### 4.2. Growth Experiments

The ciliate *Mesodinium rubrum* culture grown until the late exponential growth phase (ca. 5.0 × 10^3^ cells per mL^−1^) was diluted with the fresh culture media when it reached the late exponential growth phase to obtain an initial cell abundance of ca. 2.0 × 10^3^ cells per mL^−1^, and 7.5 mL aliquots of the mixed culture were inoculated into the wells of 6-well microplates (Iwaki, Japan). Next, 125 µL of a *D. norvegica* culture (strain DN16062021FUN-06) containing 375 cells was added into the *M. rubrum* culture to obtain an initial concentration of 50 cells per mL^−1^. The growth experiment was conducted for 36 days under the same light and temperature conditions used for maintaining the culture of *D. norvegica*. In the growth experiment, 1 mL of the cultures (triplicate) was sampled for toxin analysis every three days except for Day 33. *D. norvegia* and *Mesodinium* cell abundances were obtained as described in the previous paragraph. The specific growth rate (µ, divisions per day^−1^) of *D. norvegica* was determined during the exponential growth phase according to [95].

### 4.3. Sequences of 28S rDNA (D1-D2 Region)

Genomic DNA was extracted from several cells in each strain with 5% Chelex buffer in four of the established strains [96]. The reaction mixture for PCR amplification was prepared by adding 1 μL of template DNA, 1 μM of both D1/D2 primer sets [97], 0.2 mM of each dNTP, 1× PCR buffer, 1.5 mM Mg^2+^, 1U KOD-Plus-Ver.2 (TOYOBO, Osaka, Japan), and RNA-free dH_2_O to achieve a final volume of 25 μL. The amplification was carried out using a thermal cycler (PC-808, ASTEC, Fukuoka, Japan). PCR amplification was performed with the following cycling conditions: 2 min at 94 °C, 30 cycles at 94 °C for 15 s, 55 °C for 30 s, and 68 °C for 40 s. The PCR products were transformed into DH5α cells (Promega, Madison, WI, USA) after ligation into the pGEM T-Easy Vector (Promega). After color selection, plasmid DNA was purified and the DNA sequences were determined using M13 Reverse and U19 primers with a Dynamic ET terminator cycle sequencing kit (GE Healthcare, Little Chalfont, UK). The sequences were then analyzed on a DNA sequencer (ABI3730, Applied Biosystems, Foster City, CA, USA). A BLAST search was performed to determine closely related species, and their GenBank sequences were obtained for phylogenetic analyses. The sequences were aligned using AliView [98], and identical sequences were compiled into a single sequence. All newly obtained sequences were deposited into the DDBJ databank (accession numbers: LC760478-LC760481).

A phylogenetic tree was constructed based on maximum likelihood (ML) using MEGA version 10 [99] with the best substitution model selected: Kimura 2-parameter model plus gamma distribution (G = 0.86). Bootstrap support (BS) values of ML and neighbor-joining (NJ) analyses for the trees were estimated using 500 replicates each. For the posterior probabilities (PP) of Bayesian inference, the best model substitution calculated by Akaike information criterion in jModelTest version 2.1.10 [100] was TIM plus gamma (G = 0.9710), and the effective sample size was calculated using Bayesian Evolutionary Analysis Sampling Trees (BEAST) and Tracer. Bayesian inference was conducted using MrBayes version 3.2.5 [101] based on the Bayesian information criteria calculated by jModelTest. A total of 3,077,000 Markov chain Monte Carlo generations were used with 4 chains and trees sampled every 1000 generations with PP estimated with 25% generations burn-in. Convergence of the chains was reconfirmed when the average standard deviations of the split frequencies were below 0.01 after calculations. Sequences from *Prorocentrum micans* Ehrenberg, 1834, and *Prorocentrum cordatum* (Ostenfeld) J.D.Dodge, 1976, were used as outgroup.

### 4.4. DSP Toxin Analysis

The samples were frozen at −30 °C until the toxins were extracted by solid-phase extraction (SPE). The SPE of toxins was modified compared to a previous method [86,102,103,104]. The MonoSpin C18 centrifuge cartridge column (GL Science Inc., Tokyo, Japan) was loaded with 1 mL of the thawed samples and conditioned with 1.0 mL each of methanol and distilled water. The SPE column was rinsed with 0.5 mL distilled water, and the toxins were extracted with 0.1 mL methanol. The methanol eluates were subjected to LC-MS/MS analysis using a previously established method. [105]. A Nexera-20XR series liquid chromatograph (Shimadzu, Kyoto, Japan) was coupled to a QTRAP 4500 mass spectrometer (SCIEX, MA, USA) of a hybrid triple quadrupole/linear ion trap. Separations were performed on LC columns (internal diameter [i.d.], 100 mm × 2.1 mm) packed with 1.9 μm Hypersil GOLD C8 (Thermo Fisher Scientific Inc., Waltham, MA, USA) and maintained at 30 °C. Eluent A was composed of water, while eluent B was a mixture of 95% acetonitrile and 5% water, which contained both 2 mM ammonium formate and 50 mM formic acid. Toxin elutions were carried out from the column with 50% B at a flow rate of 0.3 mL per min^−1^. LC-MS/MS analysis was performed using multiple-reaction monitoring (MRM) with negative-mode ionization. The target parent ions and fragment ions in Q1 and Q3 were used for each toxin, as follows: OA, *m*/*z* 803.5 > 255.1; DTX1, *m*/*z* 817.5 > 255.1; PTX2, *m*/*z* 857.5 > 137.0; PTX1 and PTX11, *m*/*z* 873.5 > 137.0; PTX2 Seco acid (PTX2 SA), *m*/*z* 875.5 > 137.0. The lowest detection limits of OA/DTX1 and PTX2 were 0.1 and 1.2 ng per mL^−1^. The LC-MS/MS method was used to analyze 100 cells of the toxic plankton, which showed levels equivalent to 0.1 pg/cell of OA (and DTX1) and 1.2 pg/cell of PTX2. During the growth experiments, the specific toxin production rate (µtox, pg per cell^−1^ per day^−1^) and net toxin production rate (R_tox_) were calculated for the exponential growth phase, using the previously published equations [106].

## Figures and Tables

**Figure 1 toxins-15-00318-f001:**
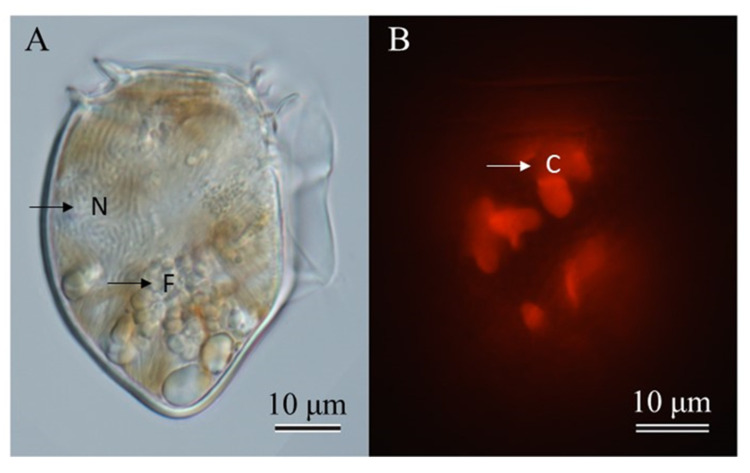
Micrographs of a vegetative cell of *Dinophysis norvegica* in culture in a bright field (**A**) and with fluorescence under blue light excitation (**B**). Plastids of *D*. *norvegica* emit red autofluoresecence. Cells in the early exponential growth phase were used. Scale bar is 10 µm, C: chloroplast, F: food vacuole, N: nucleus.

**Figure 2 toxins-15-00318-f002:**
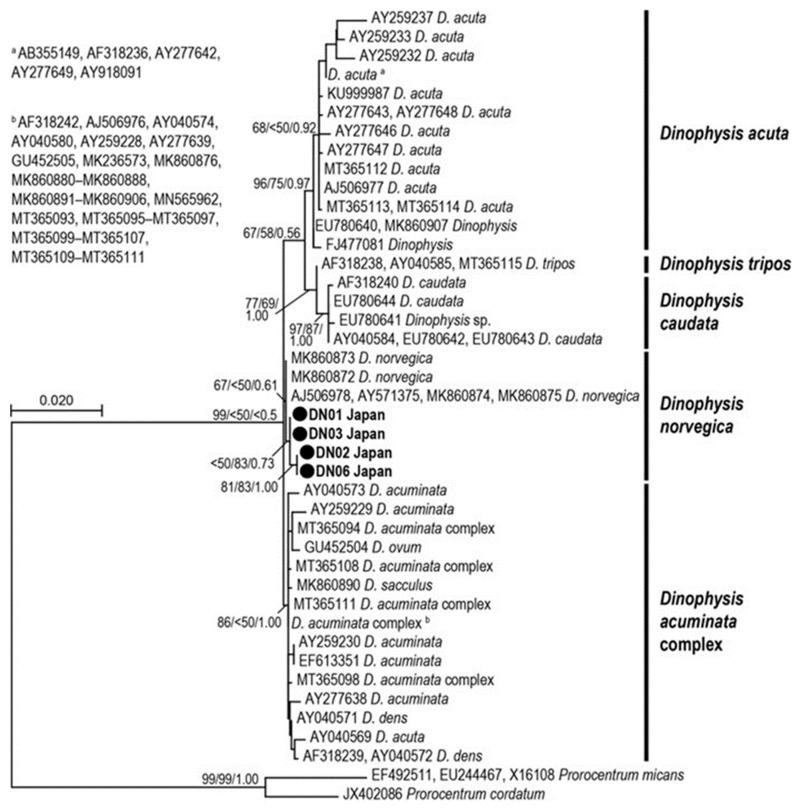
Maximum likelihood (ML) tree of *Dinophysis norvegica* inferred from D1/D2 region (735 b.p.). Bootstrap supports of ML and neighbor joining (NJ) and posterior probabilities (PP) of Bayesian inference are indicated at node (ML/NJ/PP). Culture strains obtained in this study are in bold.

**Figure 3 toxins-15-00318-f003:**
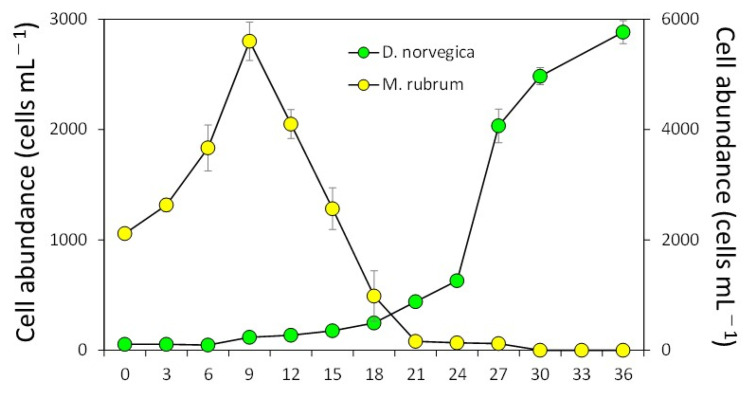
Growth of strain DN16062021FUN-06 of *Dinophyis norvegica* in culture with the ciliate *Mesodinium rubrum*.

**Figure 4 toxins-15-00318-f004:**
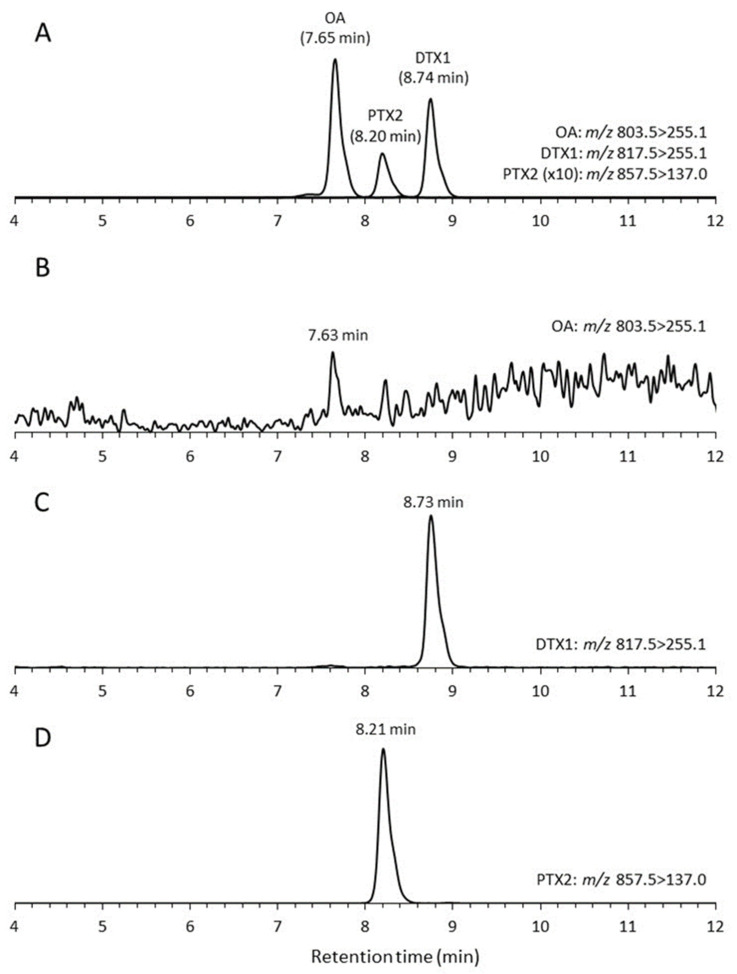
The MRM chromatograms of lipophilic shellfish toxins by LC-MS/MS analysis. (**A**): The MRM chromatograms of lipophilic shellfish toxins standards OA, DTX1, and PTX2 100 ng per mL^−1^. (**B**): A chromatogram of OA MRM transition (*m*/*z* 803.5 > 255.1) from *Dinophysis norvegica* strain DN16062021FUN-08 extract. (**C**): A chromatogram of DTX1 MRM transition (*m*/*z* 817.5 > 255.1) from strain DN16062021FUN-08 extract. (**D**): A chromatogram of PTX2 MRM transition (*m*/*z* 857.5 > 137.0) from strain DN16062021FUN-08 extract.

**Figure 5 toxins-15-00318-f005:**
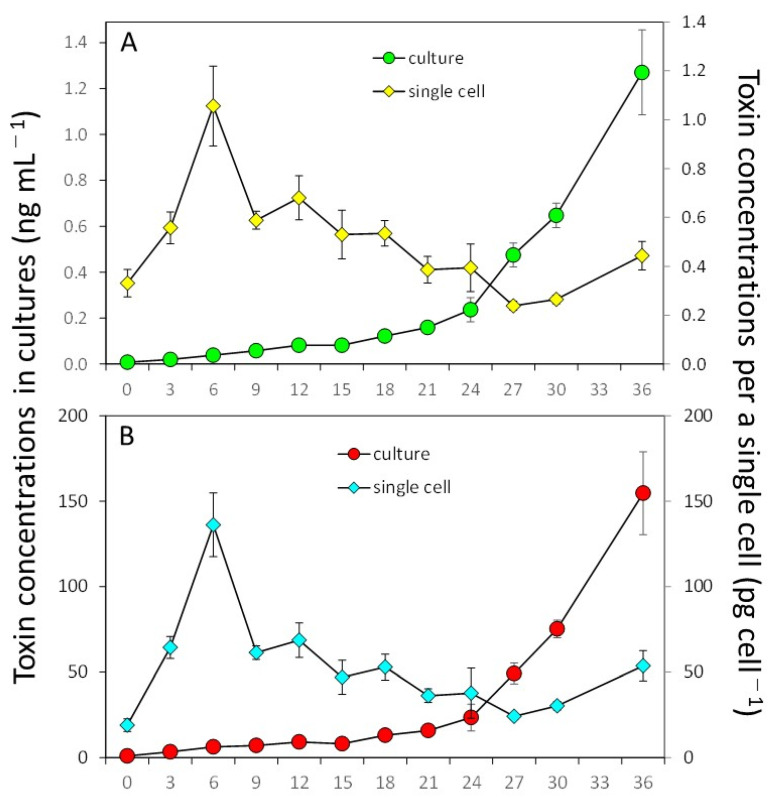
Toxin production of strain DN16062021FUN-06 of *Dinophysis norvegica* in culture with the ciliate *Mesodinium rubrum*. (**A**): Dinophysistoxin-1, (**B**): Pectenotoxin-2.

**Table 1 toxins-15-00318-t001:** Toxin production in seven cultured strains of *Dinophysis norvegica*.

Strains	Concentration in Culture (ng per mL^−1^)			Number of Cells	Cell Quota (pg per mL^−1^)		
	OA	DTX1	PTX2		OA	DTX1	PTX2
DN16062021FUN-01	ND (<0.1)	ND (<0.1)	137	1807	ND (<0.01)	ND (<0.01)	75.8
DN16062021FUN-02	ND (<0.1)	ND (<0.1)	126	2080	ND (<0.01)	ND (<0.01)	60.6
DN16062021FUN-03	ND (<0.1)	ND (<0.1)	145	2333	ND (<0.01)	ND (<0.01)	62.1
DN16062021FUN-05	ND (<0.1)	1.44	375	2850	ND (<0.01)	0.5	131.6
DN16062021FUN-06	ND (<0.1)	3.55	316	3050	ND (<0.01)	1.2	103.6
DN16062021FUN-07	ND (<0.1)	ND (<0.1)	161	1057	ND (<0.02)	ND (<0.02)	152.4
DN16062021FUN-08	ND (<0.1)	0.705	132	936	ND (<0.02)	0.7	137.0

OA: Okadaic acid; DTX1: Dinophysistoxin-1; PTX2: Pectenotoxin-2; ND: Not detected (limit of detection); After one month of incubation, 1 mL of each culture was sampled for toxin analysis.

## Data Availability

Data are contained within the article. Access to sequencing data is described in Section 4.3.

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
