# Peer review of "Growth, Toxin Content and Production of Dinophysis Norvegica in Cultured Strains Isolated from Funka Bay (Japan)"

_toxins, 2023, doi:10.3390/toxins15050318_

Round 1
Reviewer 1 Report
Abstract is very poorly written. English and format of the abstract need to be improved.
In several sections of the MS correct the format of the scientific names. At the first mention of a species give taxonomic authorities
Introduction
Line 31. Separate -to important-
Line 45: such as dinophysistoxins (DTX)….the bioactive pectentoxins (PTX)
Line 86, specify the strain
Results
Line 93. Separate -the widest-
Line 112. At the end of 36 days of incubation
Line 114. Was able to feed on M. rubrum and grow
Line 123. What to authors mean by -Consumption by D. norvegica and natural death- is this a subtitle? In the following lines they describe the growth of D norvegica, but nothing is mentioned about the consumption of M rubrum, neither natural death.
Line 126- (from day 24 to day 27)
Line 127. Every third day
Line 133. Strain
Table 1. In the title specify in which day cells were harvested for toxin analyses. Authors also need to mention this in the method section.
Figure 4. What do authors mean by -cell + medium-? correct cell quota. Add -A- and -B- to the graphs
Discussion
Line 168. Eliminate -and-…D fortii isolated from …
Line 179. had a much longer
Line 180. Eliminate -those in-
Line 182. What is the change in ph levels they undergo when added to M rubrum cultures, provide more detail
Line 189. Establishment of strains of species of Dinophysis is important….
Line 190…and understand how it is ….
Line 194. remained below…(delete mainly)
Line 197. Specify the regions from Japan and Norway where they were isolated from
Line 208. What type of organisms are M rubrum and T amphixeia? Explain to the readers briefly.
Line 240. Mesodinium rubrum was diluted with fresh culture media when it reached the late exponential growth phase to obtain an initial cell abundance of xxx….
Line 243. Delete -Next:
Line 244 to obtain
Line 248. What type of counting chamber was used?
Line 267. What does DDBJ stand for? Provide a table with the information of the sequences used for the phylogenetic tree (strain code, place of isolation, date of isolation, references, etc)
Author Response
Ms. Ref. No.: Toxins-2283001
Title: Growth, toxin content and production of Dinophysis norvegica in cultured strains isolated from Funka Bay (Japan)
Toxins
The authors are grateful to the reviewers for their valuable comments and suggestions. All issues indicated by the reviewers have been addressed and the responses are included below in italics as points under the respective comment. The changes made are highlighted as track changes in the original manuscript.
Reviewer 1
Abstract is very poorly written. English and format of the abstract need to be improved. In several sections of the MS correct the format of the scientific names. At the first mention of a species give taxonomic authorities
Thank you for the feedback, both issues have been addressed in the revised version.
>Introduction
>Line 31. Separate -to important-
Revised accordingly.
>Line 45: such as dinophysistoxins (DTX)….the bioactive pectentoxins (PTX)
Revised accordingly.
>Line 86, specify the strain
Revised accordingly.
>Results
>Line 93. Separate -the widest-
Revised accordingly.
>Line 112. At the end of 36 days of incubation
Revised accordingly.
>Line 114. Was able to feed on M. rubrum and grow
Revised accordingly.
>Line 123. What to authors mean by -Consumption by D. norvegica and natural death- is this a subtitle? In the following lines they describe the growth of D norvegica, but nothing is mentioned about the consumption of M rubrum, neither natural death.
This is not a subtitle and may be caused by miss formatting.
>Line 126- (from day 24 to day 27)
Revised accordingly.
>Line 127. Every third day
Revised accordingly.
>Line 133. Strain
Revised accordingly.
>Table 1. In the title specify in which day cells were harvested for toxin analyses. Authors also >need to mention this in the method section.
I added the incubation period at the bottom of the Table. It had already been in Line 233, “After one month of incubation, 1 mL of each culture was sampled for toxin analysis”.
>Figure 4. What do authors mean by -cell + medium-? correct cell quota. Add -A- and -B- to the graphs
We changed the usage guide from “cell + medium” to “culture” and from “cell quota” to “single cell”, and added A and B.
>Discussion
>Line 168. Eliminate -and-…D fortii isolated from …
Revised accordingly.
>Line 179. had a much longer
Revised accordingly.
>Line 180. Eliminate -those in-
Revised accordingly.
>Line 182. What is the change in ph levels they undergo when added to M rubrum cultures, provide more detail
This is just our possible speculation. Perhaps, pH changes largely before and after reinoculation. Because Mesodinium grows exponentially at the beginning, hence pH increases rapidly because of active photosynthesis. Dinophysis grows according to the consumption of Mesodinum. pH decreases gradually because of less photosynthetic activity by Medosinium and drops in pH accelerate when the ciliate disappears in the culture.
>Line 189. Establishment of strains of species of Dinophysis is important….
It was changed to “The establishment of strains in the toxic…”.
>Line 190…and understand how it is ….
Revised accordingly.
>Line 194. remained below…(delete mainly)
Revised accordingly.
>Line 197. Specify the regions from Japan and Norway where they were isolated from
Unfortunately, there seems to be a mistake in the line numbers as line 197 refers to page 9 in the first submitted version of the manuscript and contains sentences on toxin production in this study (no reference to Norway).
Maybe, line 297 was meant, where the toxin production in strains isolated from Japan and Norway was mentioned based on the previous studies. We have added the locations of the isolates reported in previous studies there:
- 273-276 “Until now, information on the concentration of toxins produced by D. norvegica has been available based on picked and pooled cells from field samples from two locations in Norway (Sogndal and Flødevigen Bay) and in Japan (Yakumo and Saroma) [19, 22, 87].”
- 279-282 “Interestingly the PTX2 maximum concentrations were much higher in the cultured strains (152 pg cell-1), than in the cells collected from the field in Yakumo Japan (89 pg cell-1) and 1.7 pg cell-1 from Flødevigen Bay in Norway [19, 22, 87].”
>Line 208. What type of organisms are M rubrum and T amphixeia? Explain to the readers briefly.
Unfortunately, we could not locate the precise location again as in the submitted version line 208 would be in Table 1 (toxin production in D. norvegica).
Perhaps line 290 was meant, so we added the information on the group of organisms form both of the species there and also to the abstract.
- 32-26 “The strains were maintained at relatively high abundance (>2,000 cells mL-1) for more than one year when fed on the ciliate Mesodinium rubrum with the addition of cryptophyte Teleaulax amphioxeia.”
- 292-294 “The ciliate Mesodinium rubrum and cryptophyte Teleaulax amphioxeia were isolated from Inokushi Bay (32.7998 N, 131.8923 E) in Oita Prefecture, Japan, at the end of February 2007 [49].”
>Line 240. Mesodinium rubrum was diluted with fresh culture media when it reached the late exponential growth phase to obtain an initial cell abundance of xxx….
Revised accordingly.
>Line 243. Delete -Next:
We did not revise this.
>Line 244 to obtain
Revised accordingly.
>Line 248. What type of counting chamber was used?
In the counting of Dinophysis and Mesodinium cells, 100 μL of cultures (triplicate) was taken and inoculated into wells of a 96-wells plate and fixed with Lugol solution (2%). Cells were counted under the inverted microscope (Nikon TE-300).
>Line 267. What does DDBJ stand for? Provide a table with the information of the sequences >used for the phylogenetic tree (strain code, place of isolation, date of isolation, references, etc)
DDBJ stands for DNA Data Bank of Japan and usually, only the acronym is used. The same applies to NCBI and ENA (two other international nucleotide databanks). We added the accession numbers registered in DDBJ in the Materials and Method sections.
Reviewer 2 Report
Review - Manuscript Number: toxins-2283001
‘Growth, toxin content and production of Dinophysis norvegica in cultured strains isolated from Japan’
This study reports morphological, molecular and toxicological data on Japanese strains of Dinophysis norvegica. This is one of the few studies reporting this kind of information on that species.
I found the paper simple but at the same time interesting. My only doubt concerns the growth experiment which was conducted on a small volume and in plate wells. Anyway information obtained adds knowledge on Dinophysis species.
Title: ‘Growth, toxin content and production of Dinophysis norvegica in cultured strains isolated from Japan’
Please put the species name in italics and I recommend to include the name of the Japanese locality of strain isolation. So, it should be:
‘Growth, toxin content and production of Dinophysis norvegica in cultured strains isolated from Funka Bay (Japan)’
Introduction
Lines 30-32 ‘Harmful Algal Blooms (HABs) are caused by several species of microalgae in freshwater, marine, and brackish environments, which may lead toimportant ecosystematic and socioeconomic impacts as well as human illnesses’ it should be:
Harmful Algal Blooms (HABs) are caused by several species of microalgae in freshwater, marine, and brackish environments. These events may lead to important impacts at ecosystem and socioeconomic level, as well as cause human illnesses.
Line 37: ‘…dinoflagellates form the majority of toxin-producing HAB’
It should be:
‘…dinoflagellates include the majority of toxin-producing HAB..’
Line 53: Dinophysis species instead of Dinophysis spp. Please verify it along the paragraph.
Results
Lines 92-95: Please explain better how to discriminate between species! It is not clear.
Lines 109-111: maybe it could be useful to introduce the seven strain codes (using the table one for example).
Lines 111-113: ‘The cultures reached maximum cell densities of 1,057-3,050 cells ml-1 (2,020 ± 702) (mean ± SD, n = 7) at end of the one-month incubation.’
It should be: ‘Maximum cell densities of cultured strains during the one-month incubation ranged from 1,057 cells ml-1 to 3,050 cells ml-1 (mean of 2,020 ± 702 cells ml-1; n =7).’
Lines 114-116: I suggest moving to the morphology paragraph.
Lines 117-128: please rephrase and refer to the strain you are showing. In addition, why the name of the strain is different from the strain name in the Table 1. Please, standardize along the manuscript.
Figure 4: maybe quata is quota, please verify.
Discussion
The discussion is a bit messy. The results should be highlighted and discussed in order of relevance.
Please rewrite better.
Methods:
I have some problem to understand the growth experiment, or better to accept such a small volume for a growth experiment (if I understand: 7,5 ml each of the 6 walls). I have not found literature where experiments of this type are done but I could have been wrong in the search. Please, if there is literature, I would cite it as a reference in the methods. Withdrawing such a large amount of volume (about 30 mL in a month if I'm not mistaken, counting three replicates) could lead to drastic volume reduction which could affect growth rates and toxin content, or not?
Line 107: 4.1. Isolation of clonal strains and establishment of clonal cultures
Maybe it should be better something like this: Isolation and establishment of clonal cultures.
Line 234: each strain
Line 238: equipped with a digital camera…
Line 244: were added
Line 282: Prorocentrum cordatum instead of P. minimum and please put the species author in the text when a species is mentioned for the first time.
Author Response
Ms. Ref. No.: Toxins-2283001
Title: Growth, toxin content and production of Dinophysis norvegica in cultured strains isolated from Funka Bay (Japan)
Toxins
The authors are grateful to the reviewers for their valuable comments and suggestions. All issues indicated by the reviewers have been addressed and the responses are included below in italics as points under the respective comment. The changes made are highlighted as track changes in the original manuscript.
Reviewer 2
This study reports morphological, molecular and toxicological data on Japanese strains of Dinophysis norvegica. This is one of the few studies reporting this kind of information on that species. I found the paper simple but at the same time interesting. My only doubt concerns the growth experiment which was conducted on a small volume and in plate wells. Anyway information obtained adds knowledge on Dinophysis species.
>Title: ‘Growth, toxin content and production of Dinophysis norvegica in cultured strains isolated from Japan’ Please put the species name in italics and I recommend to include the name of the Japanese locality of strain isolation. So, it should be:‘Growth, toxin content and production of Dinophysis norvegica in cultured strains isolated from Funka Bay (Japan)’
Revised accordingly.
>Introduction
>Lines 30-32 ‘Harmful Algal Blooms (HABs) are caused by several species of microalgae in freshwater, marine, and brackish environments, which may lead to important ecosystematic and socioeconomic impacts as well as human illnesses’ it should be:
Harmful Algal Blooms (HABs) are caused by several species of microalgae in freshwater, marine, and brackish environments. These events may lead to important impacts at ecosystem and socioeconomic level, as well as cause human illnesses.
Revised accordingly.
>Line 37: ‘…dinoflagellates form the majority of toxin-producing HAB’
>It should be: ‘…dinoflagellates include the majority of toxin-producing HAB..’
Revised accordingly.
>Line 53: Dinophysis species instead of Dinophysis spp. Please verify it along the paragraph.
Revised accordingly.
Results
>Lines 92-95: Please explain better how to discriminate between species! It is not clear.
We obtained the sequences of a partial LSU and compared them (Fig. 2).
>Lines 109-111: maybe it could be useful to introduce the seven strain codes (using the table one >for example).
The complete strain names were used in the manuscript and Table.
>Lines 111-113: ‘The cultures reached maximum cell densities of 1,057-3,050 cells ml-1 (2,020 ± 702) (mean ± SD, n = 7) at end of the one-month incubation.’
It should be: ‘Maximum cell densities of cultured strains during the one-month incubation ranged from 1,057 cells ml-1 to 3,050 cells ml-1 (mean of 2,020 ± 702 cells ml-1; n =7).’
It was revised as follows. Maximum cell densities of cultured strains during the 36 days of incubation ranged from 1,057 cells mL-1 to 3,050 cells mL-1 (mean of 2,020 ± 702 cells mL-1; n =7).
>Lines 114-116: I suggest moving to the morphology paragraph.
Moved accordingly.
>Lines 117-128: please rephrase and refer to the strain you are showing. In addition, why the name of the strain is different from the strain name in the Table 1. Please, standardize along the manuscript.
The complete strain names were used in the manuscript and Table.
>Figure 4: maybe quata is quota, please verify.
Revised accordingly.
>Discussion
>The discussion is a bit messy. The results should be highlighted and discussed in order of relevance. Please rewrite better.
We tried to improve.
Methods:
I have some problem to understand the growth experiment, or better to accept such a small volume for a growth experiment (if I understand: 7,5 ml each of the 6 walls). I have not found literature where experiments of this type are done but I could have been wrong in the search. Please, if there is literature, I would cite it as a reference in the methods. Withdrawing such a large amount of volume (about 30 mL in a month if I’m not mistaken, counting three replicates) could lead to drastic volume reduction which could affect growth rates and toxin content, or not?
In the method section, we described it as follows: and 7.5 mL aliquots of the mixed culture were inoculated into the wells of 6-well microplates (Iwaki, Japan). We used several microplates to carry out the 36 days incubation experiments. So, there was no problem practically.
>Line 107: 4.1. Isolation of clonal strains and establishment of clonal cultures
Maybe it should be better something like this: Isolation and establishment of clonal cultures.
Revised accordingly.
Line 234: each strain
This has been rephrased accordingly.
>Line 238: equipped with a digital camera…
This has been changed accordingly.
>Line 244: were added
This has been changed accordingly.
>Line 282: Prorocentrum cordatum instead of P. minimum and please put the species author in >the text when a species is mentioned for the first time.
This has been changed accordingly and the authors for the species have been added.
Reviewer 3 Report
The text and the font need to be formatted, in some places a better wording and a small English correction is needed.
Line 2: Is the species in question Dinophysis norvegica Claparède & Lachmann, 1859?
Line 5: Please rewrite the part '7 of 48 isolated cells were established as clonal cultures, 14.5%' to make the statement clearer
Line 7: Insert ‘ciliate’ before ’Mesodinium’ (also in lines 121, 161, 176, 208 and 240)
Lines 7 and 8: Please write the species names in italics (check also the rest of the manuscript); change 'for more than one year’ to ‘throughout the year’ or give the number of months (the same remark applies to line 172), as this was not further mentioned in the manuscript
Line 8: Add ‘cryptophyte’ before ’Teleaulax‘(also in lines 208 and 231)
Line 19: Please change 'in' to 'of'
Line 25 and 44: Please change 'diarrheic' to 'diarrhetic '
Line 26: Instead of 'several' please indicate the exact number or range
Line 31: What was meant by 'important ecosystematic impacts'?
Line 33: The text says 'over the past decades' while the references are from 1990, 1993, and three from 2000, so none for the last two decades; please add more recent ones
Lines 40 to 42: The references cited in this section are from 1998, 1978, and 1989, but it would be better to (also) review the literature of those from today, 40 years later, to confirm that these statements are still valid
Line 64: Change 'hardship to fisheries and aquaculture industries' to 'difficulties for the fishing and aquaculture industry’
Line 65: Add 'areas' after 'production'
Lines 71 and 72: The 78 refers to Mumbai, the 79 to Mexico, and the 80 to eastern Canada, but there is no reference for the Baltic Sea
Lines 72, 78, 81, 83 and 192: Change 'picked' to 'harvested' or 'manually harvested'
Line 72: Please mark the year for the statement 'the earliest information'
Line 73: Please list other countries from which samples were also collected and described in the paper with reference number 17; change ‘environmental’ to ‘field’ (also in lines 78 and 81)
Line 91: Change ‘Fig’ to ‘Figure’ (also in lines 97, 115, 116, 118, 128, 138, 171, and 180)
Line 100: Use either ‘micrographs’ or ‘microphotographs’ (in line 114) in the manuscript; the text here says ‘left’ and ‘right’, but micro(photo)graphs are marked A and B, please, use A and B also as in the text
Lines 101 and 102: Please, mark the organelles (nucleus, food vacuoles and chloroplasts) on the micro(photo)graphs that are supposed to be according to lines 114, 115 and 116; note the origin of these cells and date
Line 106: Delete ‘highlighted’
Line 107: Note the origin of these strains
Line 118: Delete ‘until’
Line 121: Insert the information about the origin of the strain and the experimental conditions (cultivation)
Line 123: Change ‘Consumption’ to ‘consumption’
Line 127: Maybe to change ‘in every three days’ to ‘in three-day averages’
Line 133: Change ‘stain’ to ‘strain’ (also in line 234)
Line 145: What was meant by ‘were available’ (delete)?
Line 156 to 158: Please rephrase the following part of the manuscript ‘ND: not detected (limit of detection), Trace: a signal to noise ratio was about 3 by the LC/MS/MS, but lower than the limit of detection.’, since three times the signal-to-noise ratio is the limit of detection and ten times the signal-to-noise ratio is the limit of quantification, results above LOQ can be reliably quantified, while results between LOD and LOQ can be considered trace values, while results below LOD must not be reported at all; the manuscript does not contain information on the limits of quantification of the relevant toxins
Line 159: The diagram in Figure 4 is well constructed, but the legend labelling and the titles of the diagram are confusing and need to be revised
Lines 163 to 205: Please make it clearer which part of the text refers to the present study and which part deals with the other papers
Line 167: Change ‘in’ to ‘by’
Line 168: Move ‘and’ in front of ’D. fortii‘
Line 171: Figure 2 does not contain information on temperature
Line 193 through 199: Could you explain the relevance of reference 19 to the statement in these sentences?
Lines 199 and 200: The information in this sentence ‘From 1987 until 2022, 43 HABs associated with D. norvegica have been recorded globally with the majority of those associated with DSP’ is not contained in reference 90
Line 201: Please explain the part ‘1% change in the production’
Lines 208 and 209: Does this sentence refer to this study? Are Mesodinium rubrum and Teleaulax amphioxeia maintained form 2007?
Line 213: What does ‘any’ stand for?
Line 216: Does the part ‘at the sampling site’ mean where the Dinophysis cells were sampled?
Line 231: Delete ‘for the maintenance’
Line 252: Instead of ‘several’ insert approximate number or range
Line 285: Please note the applied modification of SPE
Lines 286 to 289: Please arrange sentences in order of actions (frozen, thawed, conditioned, eluted); change ‘equlibrate’ to ‘conditioned’
Lines 289 and 290: Please delete one ‘LC-MS/MS’ and combine these two sentences into one
Line 296: Add ‘both’ before ‘containing two mM ammonium formate’
Line 457: There is no mention of reference 70 in the manuscript
Author Response
Ms. Ref. No.: Toxins-2283001
Title: Growth, toxin content and production of Dinophysis norvegica in cultured strains isolated from Funka Bay (Japan)
Toxins
The authors are grateful to the reviewers for their valuable comments and suggestions. All issues indicated by the reviewers have been addressed and the responses are included below in italics as points under the respective comment. The changes made are highlighted as track changes in the original manuscript.
Reviewer 3
>The text and the font need to be formatted, in some places a better wording and a small English correction is needed.
Thank you for indicating this. It has been addressed in the revised version of the manuscript.
>Line 2: Is the species in question Dinophysis norvegica Claparède & Lachmann, 1859?
Yes, and thank you for mentioning this. The information on the researchers’ first describing the species has been added to the text.
>Line 5: Please rewrite the part '7 of 48 isolated cells were established as clonal cultures, 14.5%' to make the statement clearer
Thank you for indicating this. This part of the sentence has been removed.
>Line 7: Insert ‘ciliate’ before ’Mesodinium’ (also in lines 121, 161, 176, 208 and 240)
Thank you for the suggestion. The word “ciliate” has been added to the suggested locations in the text.
>Lines 7 and 8: Please write the species names in italics (check also the rest of the manuscript); change 'for more than one year’ to ‘throughout the year’ or give the number of months (the same remark applies to line 172), as this was not further mentioned in the manuscript
Thank you for the feedback. Both issues have been addressed in the revised version of the manuscript.
>Line 8: Add ‘cryptophyte’ before ’Teleaulax‘(also in lines 208 and 231)
This has been modified accordingly.
>Line 19: Please change 'in' to 'of'
This has been changed accordingly.
>Line 25 and 44: Please change 'diarrheic' to 'diarrhetic '
This has been modified accordingly.
>Line 26: Instead of 'several' please indicate the exact number or range
Thank you, we have specified that the toxin production data is based on seven strains.
>Line 31: What was meant by 'important ecosystematic impacts'?
The sentence has been rephrased.
- 67-69 “These events may lead to important impacts at the ecosystem and socioeconomic levels, as well as cause human illnesses [1-3].”
>Line 33: The text says ‘over the past decades’ while the references are from 1990, 1993, and three from 2000, so none for the last two decades; please add more recent ones
Thank you, the text has been modified to fit with the references.
>Lines 40 to 42: The references cited in this section are from 1998, 1978, and 1989, but it would be better to (also) review the literature of those from today, 40 years later, to confirm that these statements are still valid
A old paper [20] was replaced to a new one.
>Line 64: Change 'hardship to fisheries and aquaculture industries' to 'difficulties for the fishing and aquaculture industry’
The sentence has been rephrased accordingly.
>Line 65: Add 'areas' after 'production'
This has been added.
>Lines 71 and 72: The 78 refers to Mumbai, the 79 to Mexico, and the 80 to eastern Canada, but there is no reference for the Baltic Sea
Thank you for indicating this, the references to the Baltic Sea have been added.
>Lines 72, 78, 81, 83 and 192: Change 'picked' to 'harvested' or 'manually harvested'
This has been changed to “isolated” throughout the manuscript.
>Line 72: Please mark the year for the statement 'the earliest information'
The year (1989) has been added to the sentence.
>Line 73: Please list other countries from which samples were also collected and described in >the paper with reference number 17; change ‘environmental’ to ‘field’ (also in lines 78 and 81)
The word “environmental” has been replaced by “field” throughout the manuscript.
>Line 91: Change ‘Fig’ to ‘Figure’ (also in lines 97, 115, 116, 118, 128, 138, 171, and 180)
This has been changed accordingly throughout the manuscript.
>Line 100: Use either ‘micrographs’ or ‘microphotographs’ (in line 114) in the manuscript; the >text here says ‘left’ and ‘right’, but micro(photo)graphs are marked A and B, please, use A and >B also as in the text
Thank you for suggesting this. The changes have been made accordingly.
>Lines 101 and 102: Please, mark the organelles (nucleus, food vacuoles and chloroplasts) on >the micro(photo)graphs that are supposed to be according to lines 114, 115 and 116; note the >origin of these cells and date
We added arrows and abbreviations for nucleus, food vacuoles, and chloroplasts ad N, F, and C, respectively in the figure caption. The following sentences were added in the text. Cells grown in the mid exponential growth phase were used. C, chloroplast; F, food vacuole, N, nucleus.
>Line 106: Delete ‘highlighted’
This has been removed.
>Line 107: Note the origin of these strains
The origin of the strains has been added (Funka Bay) to the title of subsection 2.2.
>Line 118: Delete ‘until’
This has been deleted.
>Line 121: Insert the information about the origin of the strain and the experimental conditions >(cultivation)
The information on the origin of the M. rubrum strain has been added as well as the information on experimental conditions.
- 161-162 “Only seven cultures of 48 single-cell isolates grew with the addition of the ciliate Mesodinium rubrum from the Oita Prefecture (Japan) as the prey species.”
l.169-171 “In the growth experiment (12.5 ËšC, 12:12 light:dark cycle and irradiance of 100 μmol m–2 s–1), the ciliate M. rubrum grew exponentially during the first 9 days, reaching 5,600± 346 cells mL-1 (mean ± SD, n = 3) (Figure 3).”
>Line 123: Change ‘Consumption’ to ‘consumption’
This has been modified accordingly.
>Line 127: Maybe to change ‘in every three days’ to ‘in three-day averages’
This has been rephrased to “every third day”.
>Line 133: Change ‘stain’ to ‘strain’ (also in line 234)
This has been changed accordingly.
>Line 145: What was meant by ‘were available’ (delete)?
This has been rephrased as “were detected”.
>Line 156 to 158: Please rephrase the following part of the manuscript ‘ND: not detected (limit of detection), Trace: a signal to noise ratio was about 3 by the LC/MS/MS, but lower than the limit of detection.’, since three times the signal-to-noise ratio is the limit of detection and ten times the signal-to-noise ratio is the limit of quantification, results above LOQ can be reliably quantified, while results between LOD and LOQ can be considered trace values, while results below LOD must not be reported at all; the manuscript does not contain information on the limits of quantification of the relevant toxins
We revised Table 1 according to the suggestion and added a figure (Fig. 4).
>Line 159: The diagram in Figure 4 is well constructed, but the legend labelling and the titles of the diagram are confusing and need to be revised
We changed the usage guide from “cell + medium” to “culture” and from “cell quota” to “single cell”, and added A and B.
>Lines 163 to 205: Please make it clearer which part of the text refers to the present study and which part deals with the other papers
Thank you for indicating this. The text has been revised to emphasize the work done in this study and in other studies.
>Line 167: Change ‘in’ to ‘by’
Changed accordingly.
>Line 168: Move ‘and’ in front of ’D. fortii‘
Changed accordingly.
>Line 171: Figure 2 does not contain information on temperature
Reference to Figure 2 has been removed.
>Line 193 through 199: Could you explain the relevance of reference 19 to the statement in these sentences?
Reference 19 has been removed.
>Lines 199 and 200: The information in this sentence ‘From 1987 until 2022, 43 HABs associated with D. norvegica have been recorded globally with the majority of those associated with DSP’ is not contained in reference 90
This a reference to the database, where a search with certain criteria (in this case “Dinophysis norvegica”) has to be made to access the results. It is unfortunately not possible to cite the search results directly.
>Line 201: Please explain the part ‘1% change in the production’
Thank you for indicating this. The sentence has been rephrased to convey the meaning better.
- 290-293 “A study based on Scottish shellfish farms estimated that a 1% increase in the toxin production by Dinophysis spp. can lead to a 0.66% reduction in shellfish production, resulting in an estimated annual loss of 1.37 million £ (GBP) [91].”
>Lines 208 and 209: Does this sentence refer to this study? Are Mesodinium >rubrum and Teleaulax amphioxeia maintained form 2007?
Yes, the same cultures established in 2007 were used in this study.
>Line 213: What does ‘any’ stand for?
The word “any” refers to potential nutrients present in natural seawater. The word has been replaced by “potential”.
>Line 216: Does the part ‘at the sampling site’ mean where the Dinophysis cells were sampled?
The sampling site refers to the location in Tokyo Bay from where the natural seawater was collected. The location (Tokyo Bay) has been added to the sentence to clarify this.
>Line 231: Delete ‘for the maintenance’
This has been deleted.
>Line 252: Instead of ‘several’ insert approximate number or range
This has been replaced by the number of strains (7 strains).
>Line 285: Please note the applied modification of SPE
In fact, the pressurized method was changed to the centrifugal method. So, we decided not to add the modified part in detail in the method section.
>Lines 286 to 289: Please arrange sentences in order of actions (frozen, thawed, conditioned, eluted); change ‘equlibrate’ to ‘conditioned’
Changed as follows. The 1 mL of thawed samples were applied to the MonoSpin C18 centrifuge cartridge column (GL Science Inc., Tokyo, Japan) and conditioned with 1.0-mL each methanol and distilled water.
>Lines 289 and 290: Please delete one ‘LC-MS/MS’ and combine these two sentences into one
This has been modified accordingly.
>Line 296: Add ‘both’ before ‘containing two mM ammonium formate’
This has been added.
>Line 457: There is no mention of reference 70 in the manuscript
The reference has been included in line 108.
Round 2
Reviewer 2 Report
line 369: In counting of cell abundances, 100 μL of cultures (triplicate) was taken and inoculated into wells of a 96-wells plate (Iwaki, Japan) and fixed with Lugol’s solution (2%). Cells were counted under the inverted microscope (Nikon TE-300).
should be:
Cell abundances were estimated on 100 μL of cultures (in triplicates) inoculated and fixed with Lugol’s solution (2%) into 96-wells plate (Iwaki, Japan). Cells were counted under the inverted microscope (Nikon TE-300).
line 386: The counting of cell abundances in Dinophysis and Mesodinium cells was done using the inverted microscope in the same manner as the above.
should be:
D. norvegia and Mesodinium cell abundances were obtained as described in the previous paragraph.
In addition, there are numerous typing errors in the text, please verify along the manuscript.
Author Response
Ms. Ref. No.: Toxins-2283001
Title: Growth, toxin content and production of Dinophysis norvegica in cultured strains isolated from Funka Bay (Japan)
Toxins
The authors are grateful to the reviewer for their valuable comments and suggestions. All issues indicated by the reviewer have been addressed and the responses are included below in italics as points under the respective comment. The changes made are highlighted as track changes in the original manuscript.
>line 369: In counting of cell abundances, 100 μL of cultures (triplicate) was taken and inoculated into wells of a 96-wells plate (Iwaki, Japan) and fixed with Lugol’s solution (2%). Cells were counted under the inverted microscope (Nikon TE-300).
should be:
Cell abundances were estimated on 100 μL of cultures (in triplicates) inoculated and fixed with Lugol’s solution (2%) into 96-wells plate (Iwaki, Japan). Cells were counted under the inverted microscope (Nikon TE-300).
Revised accordingly.
>line 386: The counting of cell abundances in Dinophysis and Mesodinium cells was done using the inverted microscope in the same manner as the above.
should be:
D. norvegia and Mesodinium cell abundances were obtained as described in the previous paragraph.
Revised accordingly.
>In addition, there are numerous typing errors in the text, please verify along the manuscript.
We thoroughly checked it out in the manuscript. Many thanks!